# Adipose Triglyceride Lipase in Hepatic Physiology and Pathophysiology

**DOI:** 10.3390/biom12010057

**Published:** 2021-12-31

**Authors:** Tianjiao Li, Wei Guo, Zhanxiang Zhou

**Affiliations:** 1Center for Translational Biomedical Research, North Carolina Research Campus, University of North Carolina at Greensboro, Kannapolis, NC 28081, USA; t_li4@uncg.edu (T.L.); w_guo2@uncg.edu (W.G.); 2Department of Nutrition, University of North Carolina at Greensboro, Greensboro, NC 27412, USA

**Keywords:** adipose triglyceride lipase, lipid droplets, lipolysis, liver metabolic disorder

## Abstract

The liver is extremely active in oxidizing triglycerides (TG) for energy production. An imbalance between TG synthesis and hydrolysis leads to metabolic disorders in the liver, including excessive lipid accumulation, oxidative stress, and ultimately liver damage. Adipose triglyceride lipase (ATGL) is the rate-limiting enzyme that catalyzes the first step of TG breakdown to glycerol and fatty acids. Although its role in controlling lipid homeostasis has been relatively well-studied in the adipose tissue, heart, and skeletal muscle, it remains largely unknown how and to what extent ATGL is regulated in the liver, responds to stimuli and regulators, and mediates disease progression. Therefore, in this review, we describe the current understanding of the structure–function relationship of ATGL, the molecular mechanisms of ATGL regulation at translational and post-translational levels, and—most importantly—its role in lipid and glucose homeostasis in health and disease with a focus on the liver. Advances in understanding the molecular mechanisms underlying hepatic lipid accumulation are crucial to the development of targeted therapies for treating hepatic metabolic disorders.

## 1. Introduction

The liver plays a pivotal role in regulating the metabolism of fatty acids (FA) and their neutral storage form, triglycerides (TG), which requires sophisticated coordination by multiple enzymes in lipid uptake from the circulation, de novo FA synthesis (lipogenesis), lipid usage (FA oxidations), lipid breakdown (lipolysis), and lipid secretion in the form of TG-enriched very low-density lipoprotein (VLDL-TG) [1,2]. Under physiological conditions, the liver processes large quantities of FAs with only a small portion of them converting to TG [3]. TG is further deposited into lipid droplet (LD), a spherical cellular organelle that comprises a hydrophobic core of neutral lipid and an amphipathic phospholipid monolayer [4]. However, risk factors—such as diet, alcohol, endocrine, drug, virus, and genetic variations—can induce an imbalance between lipid deposition and removal in the liver and ultimately elevate hepatocellular lipid levels. Hepatic steatosis, or fatty liver disease, is diagnosed when visible LDs are accumulated in more than 5% of hepatocytes [5]. Fatty liver can promote systemic metabolic dysfunction; progresses to steatohepatitis, fibrosis, and cirrhosis; and causes irreversible life-threatening conditions. Therefore, understanding the mechanisms regulating hepatic lipid homeostasis is critical in identifying points of intervention in this increasingly prevalent disease state. This article will focus on hepatic lipolysis and its rate-limiting enzyme, adipose triglyceride lipase (ATGL).

## 2. The Discovery of ATGL and ATGL-Mediated Lipolysis

Lipolysis is a complex and precisely regulated metabolic process for the breakdown of TG stored in cellular LDs [6]. Upon times of nutrition deprivation or enhanced energy demand, stored TG is hydrolyzed by a group of hormonally regulated hydrolytic enzymes to generate FAs and glycerol [6]. TG is initially hydrolyzed to diacylglycerol (DAG) and a molecule of fatty acid by ATGL. Next, DAG is further broken down into monoacylglycerol (MAG) and a second fatty acid by hormone-sensitive lipase (HSL). At last, MAG is converted into glycerol and a third fatty acid by monoacylglycerol lipase (MAL).

For a long time, HSL was believed as the rate-limiting enzyme in TG breakdown. However, the nonobese phenotype and accumulation of DAG in adipose tissue of HSL-deficient mice suggested that one or more hydrolases other than HSL may exist that preferentially catalyze the release of the first fatty acid from TG [7,8]. In 2004, Zimmermann et al. discovered an enzyme that has homological structures to known lipases and TG-hydrolase activity by screening the gene and protein databases and named it ATGL [9]. They reported that ATGL exhibited high specificity towards TG as the accumulation of intracellular DAGs was increased by 21-fold in the absence of this hydrolase. In contrast to HSL, ATGL does not hydrolyze cholesteryl or retinyl ester bonds [8]. In the same year, Jenkins et al. reported that calcium-independent phospholipase A2-zeta (iPLA2_zeta_) had acylglycerol transacylase activity, even though not as strong as its TG hydrolase activity [10]. Meanwhile, a third group identified a gene termed desnutrin, which encodes a protein that was believed to be a lipase. This lipase is capable of releasing FAs from adipose tissue for oxidation and is regulated by nutritional conditions, such as fasting [11]. Later, ATGL, iPLA2_zeta_, and desnutrin were collectively referred to as ATGL. Since then, emerging studies have been dedicated to understanding the physiological and pathophysiological functions and metabolic implications of ATGL as the rate-limiting enzyme in lipolysis.

## 3. The Structure–Function Relationship of ATGL

The gene of human ATGL is located on chromosome 11p15.5 and includes 10 exons that encodes a 504-amino acid protein (NP_065109) [12]. The murine gene of ATGL encodes a 486-amino acid protein (BAC27476) with a calculated molecular mass of 54 kDa, which shares an 86.8% sequence identity with its human orthologue [9,12]. Sequence analysis revealed that a patatin-like phospholipase domain between Ile 10-Leu 178 is located on the N-terminal half of the protein (Figure 1), which groups ATGL into the patatin-like phospholipase domain-containing family (PNPLA) [13]. PNPLA family members play important roles in lipid hydrolysis with varying substrate specificities—including TG, retinol ester, and phospholipid—through a catalytic dyad within a three-layer (α-β-α) sandwich structure. Mutation studies confirmed that certain residues, such as Ser47 and Asp 166, within the patatin-like containing domain constitute the putative catalytic dyad in ATGL (Figure 1) [14,15,16]. ATGL is synthesized in the endoplasmic reticulum (ER) and directly delivered to LDs by the Golgi-ER transport protein complex GBF1 (Golgi-specific brefeldin A resistance factor 1)-Arf1 (ADP-ribosylation factor 1)-COPI (coat protein complex I) [17]. It has been reported that ATGL traffics from ER to LDs through membrane bridges that are controlled by the Arf1/COPI machinery [17]. ATGL is localized in cytoplasm, on LDs, and on plasma membranes [13].

The studies of naturally occurring mutations in the human ATGL gene in patients with neutral lipid storage disease with myopathy (NLSD-M) as well as in experimental models of ATGL mutations, have greatly advanced our knowledge of the structure–function relationship of ATGL and its role in health and diseases. The N-terminal region of ATGL, containing the patatin-like domain and a GXSXG consensus motif with the active serine, is believed to be responsible for ATGL’s TG hydrolase activity based on the analysis of C-terminally truncated ATGL found in patients with NLSD-M [14,18,19]. Cell culture studies have confirmed that both active sites, Ser47 and Asp166, are indispensable for TG hydrolysis [14,16]. The discovery of two overlapping glycine-rich motifs that contains an amphipathic α-helix structure within residues 10–24 in the proximal N-terminal suggested that this domain may potentially have a role in TG substrate binding, as the amphipathic α-helix is involved in neutral lipid binding and is found in other enzymes with hydrolytic ability, such as a hepatic TG lipase called TG hydrolase [14].

The C-terminal region of ATGL, harboring the putative hydrophobic lipid-binding stretch (Val315 to Ile364) and the two potential AMPK phosphorylation sites (Ser 404 and Ser 428) has been demonstrated to be responsible for the localization of the enzyme around LDs [13,20,21,22]. Smirnova et al. reported that ATGL with an S47A mutation failed to decrease LD size but was still able to localize to LDs, which indicated the lack of hydrolase activity but the presence of LD-binding ability in the mutant [18]. Fisher et al. reported that the C-terminals of three ATGL mutations identified in patients with NLSD-M were severely altered and deleted, while the N-terminals were intact. In vitro, the total activity of neutral lipase in NLSD-M fibroblasts was similar to that in the control fibroblasts, suggesting that the truncated C-terminal parts are not associated with TG hydrolase activity. Later on, a study conducted by Kobayashi et al. showed that C-terminal truncation-caused mutations of ATGL failed to localize around and degrade LDs despite normal TG hydrolase activity [20].

Schweiger and coworkers confirmed that C-terminal truncated ATGL mutants do not localize to LDs, however, the TG hydrolase activities of these mutants were remarkably increased by up to 20-fold compared with wild type ATGL. The authors proposed that the C-terminal region of ATGL suppresses its TG hydrolase activity by interfering with protein–protein interaction between ATGL and its activator comparative gene identification-58 (CGI-58) [21]. Cornaciu et al. revealed that the minimal protein length for ATGL activity ranges until Leucine 254, as fragments shorter than ATGL254 failed to fulfill its TG hydrolase activity [13].

## 4. The Regulatory Mechanisms of ATGL

ATGL is expressed in all tissues with the highest abundance in white and brown adipose tissues. It is expressed at much lower levels in non-adipose tissues, such as the liver [9,23]. The central role of ATGL in lipolysis makes its regulation crucial for maintaining a defined balance between lipid storage and breakdown. The expression and activity of ATGL are regulated at multiple levels, such as transcriptional and post-translational regulations. Pharmacological activators and inhibitors of ATGL are also reviewed in this section.

### 4.1. Regulation of ATGL Expression at Transcriptional Level

The factors involved in transcriptional regulation of ATGL are summarized in Figure 2. As a hormone-sensitive lipase, ATGL is activated by β-adrenergic activation [24]. A study using mouse model revealed that AMP-activated protein kinase (AMPK) could phosphorylate ATGL at Ser406, therefore activating its enzymatic activity [25]. In contrast to HSL, phosphorylation of ATGL does not appear to involve protein kinase A (PKA) [26].

Insulin is a classic inhibitor of lipolysis. The expression of ATGL is directly suppressed in the presence of insulin through transcriptional regulation [27,28,29]. Studies revealed that ATGL expression was directly suppressed in the presence of insulin by restraining the nuclear localization of forkhead box protein O1 (FoxO1) in adipocytes. Furthermore, it was demonstrated that the FoxO1-ATGL pathway was controlled by sirtuin 1 (SIRT1) through deacetylating of FoxO1 [30]. Chakrabarti et al. firstly reported that the activation of mechanistic target of rapamycin complex 1 (mTORC1) inhibited ATGL expression at the transcriptional level upon insulin stimulation in adipocytes [31]. Later on, they further revealed that it was early growth response 1 (Egr1)—one of the early growth response transcription factor family members—that mediated mTORC1-induced downregulation of ATGL [29].

Through investigating the murine ATGL promoter, Kim et al. reported that the master adipogenic transcriptional regulator, peroxisome proliferator-activated receptor gamma (PPARγ), transactivated ATGL upon insulin stimulation [27]. Furthermore, Roy D. et al. speculated that upregulation of ATGL transcription was due to the binding of PPARγ/RXRα heterodimer in the ATGL promoter at −2424, −1674, and −1573 bp sites [32]. Insulin responsive transcription factor specificity protein 1 (Sp1) negatively regulates ATGL expression in preadipocytes. However, the physical functional interaction between PPARγ and Sp1 is necessary for the transactivation of ATGL in mature adipocytes, which indicates that PPARγ and Sp1 coordinately regulate ATGL in a state-dependent manner [27,32]. Besides, tumor necrosis factor-alpha (TNF-α) has been shown to stimulate lipolysis, partially through degrading ATGL’s direct inhibitor G0/G1 switch gene 2 (G0S2). Interestingly, TNF-α significantly downregulates ATGL mRNA levels without altering its protein abundance [32,33,34]. The underlying mechanisms by which TNF-α downregulates ATGL yet still enhances lipolysis remain to be elucidated.

### 4.2. Regulation of ATGL at Post-Transcriptional Level (Protein–Protein Interaction)

Figure 3 illustrates the regulation of ATGL at post-translational levels with a focus on the liver. In 2006, Lass and coworkers identified CGI-58 as the co-activator of ATGL, which increases the TG hydrolase activity of ATGL by up to 20-fold [15]. Since then, numerous studies have been conducted to elaborate the mechanistic link between CGI-58 and ATGL [21,35]. The α/β-hydrolase core domains within CGI-58 group it into the α/β-hydrolase domain family, which explains why it is also referred to as alpha-beta hydrolase domain-containing 5 (ABHD5). In 2017, Sanders et al. demonstrated for the first time that the two highly conserved amino acids Arg 299 and Gly 328 in CGI-58 are necessary for ATGL activation [36]. In adipose tissue, under physiological condition, CGI-58 binds to LD coating protein perilipin 1 (Plin1), thus preventing activation of ATGL. Upon PKA-mediated phosphorylation of CGI-58 and Plin1, CGI-58 is released and subsequently co-activates ATGL activity [12,37,38]. In addition, CGI-58 also localizes to the surface of LD to stimulate ATGL activity. The hydrophobic Trp-rich stretch within the N-terminal region of CGI-58 is believed to be essential for its LD localization.

G0S2 was first identified in the early 1990s in cultured blood mononuclear cells responding to drug-induced cell cycle transition from the G0 to G1 phase [39,40]. It is ubiquitously expressed in metabolically active tissues, such as adipose tissue and the liver [41]. G0S2 is known as an endogenous inhibitor of ATGL even when CGI-58 is present [42,43]. Studies have shown that global G0S2 ablation in mice enhanced adipose tissue lipolysis but decreased body weight and hepatic TG content [44,45]. Conversely, liver-specific overexpression of G0S2 in mice led to hepatic steatosis and reduced lipolysis [44]. In line with that, adipose tissue-specific overexpression of G0S2 in quails prevented fat loss under feeding-restricted conditions by inhibiting adipose tissue lipolysis [46].

As the most abundant proteins on the surface of LD in adipocytes, Perilipin family members are actively involved in regulating ATGL activity in a tissue-specific manner [47]. This family consists of five structurally related proteins, serving as the barriers of LDs and preventing lipases from getting access to them [12]. Plin1 is predominantly expressed in adipose tissue. Upon hormonal stimulation, Plin1 is phosphorylated by PKA and releases CGI-58, which subsequently co-activates ATGL [47]. Perilipin 2 (Plin2) and perilipin 3 (Plin3) are essentially expressed in all tissues, with the highest levels in the liver and small intestine [48,49]. However, they do not interact with CGI-58 due to the lack of a C-terminal part required for protein–protein interaction [50]. Studies have shown that Plin2 and Plin3 promote lipolysis through chaperone-mediated autophagy. Briefly, heat shock protein HSPA8/hsc70 mediates the degradation of Plin2 and Plin3 in lysosome and causes increased exposure of LDs surface to ATGL, thereby initiating the first step of TG hydrolysis [49,51]. It is worth noting that a recent study revealed that autophagy not only works in parallel with ATGL to facility LD degradation, but also works as a downstream target of ATGL to promote bulk LD catabolism through the mediation of SIRT1 [52]. Perilipin 5 (Plin5) is mostly expressed in oxidative tissues, such as the liver and heart, but at a lower level in adipose tissue [53,54]. In the liver, unlike Plin1 and Plin2, Plin5 could directly bind to ATGL. Studies have demonstrated that Plin5 serves as a negative regulator of ATGL-mediated lipolysis through competing against ATGL for CGI-58 interaction unless PKA is activated [54,55,56].

Hypoxia-inducible lipid droplet-associated protein (HILPDA; also known as hypoxia-inducible gene-2) is another regulator that can directly inhibit ATGL activity [57]. HILPDA is expressed in most organs, including the liver, brain, endocrine tissues, skeletal muscle, heart, lung, and adipose tissue [57,58]. HILPDA coats the surface of LDs and shares 13.5% sequence identity with G0S2, mostly in the area where G0S2 is known to bind and inhibit ATGL [59]. Studies have shown that HILPDA ablation remarkably reduced TG content and LD size in mouse primary hepatocytes [59], whereas overexpression of HILPDA increased neutral lipid deposition in HeLa cells and augmented hepatic TG content by up to 4-fold in mice [57,60]. The hydrophobic region of HILPDA within the N-terminal is believed to contribute to the intracellular TG regulation by physically interacting with the patatin-domain-containing region of ATGL, thereby inhibiting its TG hydrolase activity despite the presence of CGI-58 [61].

The ubiquitin-proteasome system also plays an essential role in ATGL modulation. Gosh et al. identified hepatic ATGL as a novel target of E3 ubiquitin ligase COP1/RFWD2 through interacting with the consensus VP motif of ATGL and targeting it for proteasomal degradation predominantly at Lys 100 residue [62]. Other protein interaction partners of ATGL have not been fully understood yet, including cell death inducing DEFA like effector C (CIDEC; also known as fat-specific protein 27, FSP27) and pigment epithelium-derived factor (PEDF; also known as seeping family F member 1, SERPINF1) [63,64,65].

### 4.3. Regulation of ATGL by Small Molecules

Atglistatin is a synthetic inhibitor of ATGL that selectively inhibits the activity of ATGL in a competitive manner [66]. Mayer et al. reported that short-term oral gavage of Atglistatin to mice markedly reduced plasma TG and the lipolytic parameters FA and glycerol without inducing TG accumulation in all tissues investigated except liver. However, it did not significantly affect blood glucose, total cholesterol, ketone bodies, and insulin levels. Atglistatin distributes differently among various tissues with the highest concentration in the liver, followed by white and brown adipose tissues, which might explain TG accumulation in the liver after administration of Atglistatin. However, Atglistatin failed to inhibit either ATGL activity or FAs release in human adipocyte [66].

G0S2 is a powerful endogenous ATGL inhibitor that is highly conservative between human and mouse [41]. Hence, G0S2-derived peptide can be used as a viable option for ATGL inhibition. Cerk et al. generated a G0S2-derived 34-amino acid peptide, hGW2052, which is a potent noncompetitive ATGL inhibitor both in mouse and human. hGW2052 is highly selective as it does not inhibit other lipases. This peptide inhibitor has great potential in translating into therapeutic application owing to its ability to fuse with other sequences to allow tissue-specific inhibition of ATGL [67].

Oroxylin A is a natural bioactive flavonoid extracted from the root of *Scutellaria baicalensis Georgi* [68]. Studies reported that Oroxylin A has a positive effect on preventing hepatic steatosis [69], ameliorating hepatic fibrosis [69,70], and promoting liver regeneration [71]. In 2019, Zhang et al. proposed that Oroxylin A restored LD contents in activated hepatic stellate cells via downregulating ATGL, therefore preventing hepatic fibrosis [72].

### 4.4. Regulation of ATGL by Lipid Intermediates

Although the regulation of ATGL activity, either by endogenous regulatory proteins or pharmacological modulators, has been excessively studied, little research has been done in exploring the role of lipid intermediates in ATGL-catalyzed lipolysis [73]. As a lipid intermediate, long-chain acyl-coenzyme A (LC-CoA) is not only a short-lived metabolite, but also a potent lipase inhibitor that inhibits lipase activity, such as HSL [73]. Nagy et al. reported that ATGL activity is inhibited by long-chain acyl-coenzyme A (LC-CoA) in a non-competitive manner regardless of the presence of CGI-58 [74]. A study showed that the inhibitory effect of LC-CoA on HSL is enhanced by acyl-CoA-binding protein (ACBP) [73]; however, whether ACBP or other acyl-CoA-binding proteins are involved in LC-CoA-mediated ATGL inhibition is unknown so far.

## 5. The Roles of Hepatic ATGL in Health and Disease

Emerging evidence have suggested that the physiological function of ATGL is not restricted to adipose tissue but is also crucially important in many nonadipose tissues, such as the liver. ATGL is expressed at low levels in hepatocytes, hepatic stellate cells (HSC), and macrophages (Kupffer cells) [9,75,76]. Hepatic ATGL levels are upregulated upon fasting [77]. Patients with NAFLD have reduced hepatic ATGL levels, although the pathophysiological mechanisms remain unclear [78]. Furthermore, in recent years, the generation of transgenic mice with global- and/or tissue-specific ATGL ablation and cellular models of ATGL manipulations, as well as patients with ATGL-related mutations, have deepened our understanding about the pivotal role of ATGL in regulating hepatic lipid homeostasis and the progression of liver diseases [21,79,80,81,82]. Here, we summarized the studies of mouse models with global- and liver-specific mutation of ATGL in Table 1 and discuss the findings in several aspects to elucidate the tissue-specific role of hepatic ATGL in the liver under normal and disease conditions (Figure 4).

### 5.1. Lessons Learned from Humans with ATGL-Related Mutations

Neutral lipid storage disease (NLSD) with myopathy (NLSD-M) is a rare autosomal recessive disorder caused by ATGL/PNPLA2 mutations. Studies investigating the clinical, molecular, and cellular phenotypes in these patients illustrate an essential role of ATGL in maintaining systemic lipid homeostasis and multi-organ functions, including the liver [19,90]. Mutations in ATGL lead to either its inactivation or compromised lipid droplet binding ability [21]. To date, 57 patients had been clinically and genetically characterized worldwide [83,85]. Thirty-nine different types of ATGL mutation had been reported that attribute to NLSD-M development by differentially affecting ATGL protein expression and/or functions [79]. NLSD-M is characterized by excessive, non-lysosomal accumulation of TG-rich cytosolic LDs in non-adipocytes, including hepatocytes [79,80]. Patients suffering from NLSD-M display a range of clinical features, such as progressive myopathy, cardiomyopathy, hepatomegaly, and diabetes with no obesity being reported [81,82].

Liver damage was observed in half of NLSD-M patients, manifesting as mild to severe hepatic steatosis and elevated blood aminotransferase levels, including alanine aminotransferase (ALT) and aspartate aminotransferase (AST) [79,83,85]. The liver biopsy results of a 63-year-old female patient with NLSD-M revealed significant hepatic cytosolic lipid deposition, indicating impaired hepatic TG turnover [21]. A 33-year-old Turkish patient with NLSD-M presented Grade 2 macrovesicular steatosis and hepatomegaly, which contributed to the unfortunate death of the patient [83].

Interestingly, another subgroup of neutral lipid storage disease: NLSD with ichthyosis (NLSD-I), caused by mutations in the *Cgi-58* gene is characterized by systemic TG accumulation and severe ichthyosis [79,82]. It is noteworthy that the liver is the most frequently damaged organ in patients with NLSD-I, the manifestations of which include hepatomegaly, steatosis, and sometimes cirrhosis [79,82]. More than 80% of NLSD-I patients have fatty liver disease [79]. A retrospective cross-sectional study conducted in Italy showed that, in a cohort of 21 patients, NLSD-I exhibited a worse prognosis in terms of life expectancy, with two deaths from hepatic failure due to lipid infiltration compared to NLSD-M [82].

### 5.2. ATGL in Hepatic TG Accumulation

It has been reported that supraphysiological TG accumulation occurred in most tissues in global ATGL deficient mice (Atgl^-/-^) compared with wild type (WT) mice [86]. Turpin et al. revealed that overproduction of ATGL markedly reduced the size of LDs in mouse liver [86]. In line with that, Reid et al. reported that only very small portions of LDs were presented in the livers of ATGL overexpressed *ob/ob* mice compared to control mice [87]. In 2006, Haemmerle et al. reported that hepatic TG content was increased by 1.3-fold in non-fasted 12- to 14-week-old male Atgl^-/-^ mice compared with WT mice [91]. Progressive hepatic steatosis was also observed in hepatocyte-specific ATGL knockout mice (Atgl^L-KO^). In 2011, Wu et al. reported that, in a time period of 12 months, Atgl^L-KO^ mice had larger cytoplasmic LDs in cholangiocytes and more than 3-fold higher TG contents compared with controls [92].

In accordance with the accumulated hepatic TG contents, the lipolytic activities in the livers of Atgl^L-KO^ mice were decreased by 33% to 73% in comparison with that in WT mice [87,91]. In an in vitro model, the lipolytic activity of ATGL was increased by 1.9-fold, while the cellular mass was reduced by about 60% in McA-RH7777 cells with adenovirus-mediated ATGL overexpression compared with that in controls [87].

To further investigate the TG-hydrolytic ability of ATGL in the liver, Reid and coworkers generated a transgenic mouse model with adenovirus-mediated hepatic overexpression of ATGL (Ad-ATGL). They reported that liver TG content was reduced by 65% in fasting female *ob/ob* mice 8 days after Ad-ATGL infection compared with Ad-GFP-infected controls. The same trend was observed in male diet-induced obesity (DIO) mice as there was a 40% reduction in hepatic TG content at 10 days post-Ad-ATGL infection compared with their counterpart controls [87]. In agreement with the aforementioned findings, Ong et al. reported that the TG hydrolase activity in the livers of mice with adenovirus-mediated hepatic ATGL knockdown was reduced by 40% after 7 days of adenovirus injection, which led to a 30% increase in liver weight and a more than doubled increase in TG content of controls [89]. Interestingly, composition analysis showed that ATGL knockdown caused a significant reduction in C16:0, C18:0, and C18:3 FAs but an increase in C18:1 FA in hepatic TG content.

Taken together, current evidence suggests that hepatic TG content is inversely correlated with the expression of ATGL in the liver. ATGL expression not only affects hepatic TG content, but also might alter the overall fatty acids profile in the liver.

### 5.3. ATGL in Hepatic FA Oxidation

PPARα is a ligand-activated transcriptional regulator of genes involved in mitochondrial FA β-oxidation [93]. Studies have shown that FAs are strong endogenous ligands that bind to and activate PPARα [94]. Given the fact that ATGL regulates the production of TG-derived FAs, it is assumed that ATGL might affect PPARα expression and therefore influence its downstream target genes that are involved in FA β-oxidation. Wu et al. reported that liver-specific knockout of ATGL in mice remarkably downregulated hepatic expressions of PPARα and carnitine palmitoyltransferase-1α (CPT-1α) at 4-month-old age compared with controls, suggesting reduced FA β-oxidation in the liver [89]. Hepatocyte-specific ATGL deficiency-induced downregulation of hepatic PPARα and its target genes was demonstrated in another adenovirus-mediated ATGL knockdown mouse model [89]. However, PPARα agonist fenofibrate treatment failed to normalize either liver weight or high-fat diet-induced elevated liver TG content in ATGL shRNA-treated male mice, suggesting that ATGL regulates PPARα through a ligand-independent manner in mice. Interestingly, in another study, Jha et al. reported that fenofibrate supplementation completely normalized methionine-choline-deficient (MCD)-induced steatosis in female ATGL-KO mice [95]. The discrepancies of how ATGL deficiency mice response to fenofibrate supplementation could be due to the various doses of fenofibrate, fasting status when mice were sacrificed, as well as the different mechanisms underlying hepatic steatosis in two animal models. Additionally, gender differences could also be a contributor to the conflicting findings, since ATGL expression has been reported to be affected by estrogen [88].

Even though to what extent that PPARα signaling pathway contributes to ATGL KO-induced steatosis is still obscure, the results about the effects of ATGL on β-oxidation are, so far, consistent. In an in vitro study, adenovirus-mediated ATGL knockdown in primary hepatocytes led to an approximately 70% decrease in fatty acid oxidation represented by ASM production, while ATGL overexpression increased ASM production by 3-folds, compared with the cells treated with control shRNA [89]. Hepatic overexpression of ATGL increased β-oxidation in mice as well as oleate supplementation-induced TG accumulation in McA-RH7777 cells [87]. As for the underlying mechanism of ATGL-derived FAs metabolism in the liver, Ong et al. demonstrated that liver fatty acid binding protein (L-FABP) is not necessary for channeling ATGL-derived FAs to mitochondria for β-oxidation or to nucleus for PPAR-α activation [84].

### 5.4. ATGL in Hepatic Inflammation

Lipotoxicity occurs when excess lipids accumulate in the liver, which may lead to hepatic inflammation. It is clinically important to investigate the role of ATGL in the pathogenesis of hepatic inflammation, which is one of the most important characteristics in both non-alcoholic fatty liver disease (NAFLD) and alcohol related fatty liver disease (AFLD) [96].

Jha et al. reported that ATGL KO mice fed with a 2-week MCD diet developed more severe inflammation in the liver compared to their WT counterparts, as evidenced by increased expression of inflammatory markers, such as TNF-α, inducible nitric oxide synthase (iNOS), monocyte chemotactic protein-1 (MCP-1), and interleukin-1beta (IL-1β), as well as increased infiltration of mononuclear inflammatory cells in the liver. For the LPS model, 8 to 10-week-old female mice were intraperitoneally injected with a single dose of LPS for 12 h at non-fasted state. Compared with WT mice, Atgl^-/-^ mice exhibited marked hepatic inflammation, as indicated by the significant upregulation of TNF-α, iNOS, MCP-1, and IL-6, along with elevated serum levels of TNF-α and IL-6. They further demonstrated that the anti-inflammatory effect of ATGL in the liver was partially achieved through PPARα signaling pathway [95].

Blood ALT levels have been used as a surrogate marker for liver injuries [97]. Wu et al. reported that Atgl^L-KO^ mice had higher plasma ALT levels and ALT/AST ratio but no significant differences in hepatic macrophage infiltration, at 8 and 12 months, compared with WT controls [92]. It is worth noting that the absence of ATGL protected mice from tunicamycin-induced acute hepatic ER stress [98].

Given the current evidence, ATGL plays a protective role against hepatic inflammation in steatohepatitis. More studies using liver-specific ATGL knockout animal models are needed to further elucidate the clinical relevance of ATGL as a therapeutic target in the progression of metabolic liver diseases.

### 5.5. ATGL in Hepatic Glucose Metabolism

Despite causing abnormal TG accumulation in non-adipose tissues, global ATGL KO was reported to be beneficial as it improved systemic glucose tolerance and insulin sensitivity [91,99,100]. The inability of TG mobilization and the subsequent reduced circulating FFAs forced the mice to shift their energy source from FFAs to glucose, therefore improving glucose clearance [91,99].

Studies have demonstrated that global ATGL deficiency promoted tissue-specific changes in insulin action [99,100]. Kienesberger et al. reported that insulin action was impaired in the liver of Atgl^-/-^ mice (at 10–12 weeks of age) in a short term fasting of 6 h, as reflected by decreased hepatic expression of insulin-stimulated phosphorylation of IRS1^Tyr612^ and Akt^Tyr308^, as well as decreased insulin-stimulated Akt activity compared with WT mice [99]. However, in an adenovirus-mediated ATGL knockdown mouse model, Ong et al. reported that hepatic insulin signaling was not altered despite improved whole-body glucose tolerance, as no changes of insulin-stimulated phosphorylation of Akt^Ser473^ or IRS1^Tyr989^ were observed in ATGL knockdown mice compared to WT mice [100]. In line with Ong, Turpin et al. reported that ATGL deficiency did not influence insulin sensitivity in hepatocytes. The insulin-stimulated phosphorylation of Akt^S473^ and Akt^T308^ was not different between genotypes, neither were the mRNA expressions of phosphoenolpyruvate kinase (*Pepck*) and glucose-6-phosphatase (*G6Pase*) upon insulin treatment. The authors claimed that ATGL ablation does not influence insulin sensitivity in hepatocytes despite induced marked steatosis, meanwhile, hepatic ATGL overexpression mildly improved hepatic insulin sensitivity without affecting hepatic inflammation [86]. These findings indicate that the improved whole-body insulin sensitivity and glucose homeostasis in Atgl^-/-^ mice may not be contributed by liver-specific insulin action.

### 5.6. ATGL in Hepatocellular Carcinoma (HCC)

During the last few years, growing evidence suggests a non-canonical role of ATGL in the prognosis of several human malignancies including HCC [101,102,103,104], yet the results are contradictory. Metabolic reprogramming is an important cancer hallmark to sustain high energy demand and fast proliferation rate via upregulation of glycolysis and lipid metabolism [105]. Several studies have reported that abnormal lipid metabolism is closely related to tumor occurrence, development, and metastasis [106,107]. Although the function of ATGL in hepatic lipid homeostasis is well-known [89], evidence regarding the role of ATGL in liver cancer is still lacking. Liu et al. have recently shown that ATGL promotes the proliferation of hepatocellular carcinoma cells by upregulating the phosphorylation of AKT, whereas inhibition of p-AKT significantly suppressed such effect mediated by ATGL [108]. Interestingly, Di Leo et al. observed that ATGL overexpression attenuated cellular glucose uptake/utilization and cell proliferation while fatty acid oxidation and mitochondria activity were enhanced. They further demonstrated that ATGL-mediated p53 acetylation/stabilization via the PPAR-α/p300 pathway is responsible for the metabolic shift from glycolysis to fatty acid oxidation [103]. In contrast, another group reported that nuclear paraspeckle assembly transcript 1 (NEAT1) disrupts HCC cell lipolysis through modulation of ATGL [109]. They found that suppression of NEAT1 in hepatoma cells downregulated the expression of ATGL, and subsequently decreased levels of cellular DAG and FFA, which led to the suppression of HCC cell proliferation. These results have provided new insights of ATGL regarding its role in HCC progression and may shed light on the development of selective therapies towards HCC with further research.

## 6. Conclusions

Although ATGL was initially characterized and well-studied in adipose tissue, growing evidence also suggests that ATGL has pivotal tissue-specific roles in other organs, including the liver. This review provides a unique perspective on the current understanding of the structure–function relationship of ATGL, the molecular mechanisms of ATGL regulation at translational and post-translational levels (summarized in Figure 2 and Figure 3), and, most importantly, its role in lipid and glucose homeostasis in health and disease (summarized in Figure 4). Many important questions regarding hepatic ATGL and ATGL-mediated lipid metabolism remain unanswered, including: (a) How is hepatic ATGL regulated in the setting of fatty liver diseases? (b) What is the precise mechanism by which hepatic ATGL mediates the onset and progression of fatty liver diseases? (c) Do cell–cell interactions among hepatocytes, HSCs, and macrophages exist when hepatic ATGL is dysregulated? and (d) Is hepatic ATGL of clinical relevance and should it be targeted as a potential therapeutic strategy for liver diseases, including fatty liver disease, steatohepatitis, hepatic fibrosis, or even liver cancer? In summary, a deeper understanding is needed regarding the precise mechanisms whereby FA uptake, synthesis, and breakdown are tightly regulated within the liver, especially through the regulation of hepatic ATGL.

## Figures and Tables

**Figure 1 biomolecules-12-00057-f001:**
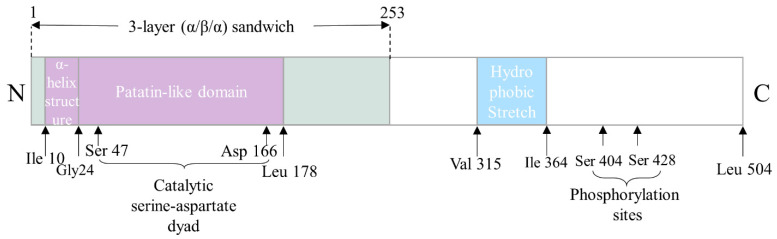
Schematic representation of predicted structure and domain organizations of hATGL. Green: three-layer (α/β/α) sandwich domain (residue 1–253). Purple: Patatin-like domain (residue 10–178), including the α-helix structure (residue 10–24) and the catalytic serine-aspartate dyad (Ser 47 and Asp 166), which are essential in TG substrate binding and TG hydrolysis, respectively. Blue: putative hydrophobic lipid-binding stretch (residue 315–364), and two potential AMPK phosphorylation sites (Ser 404 and Ser 428), which are responsible for the localization of ATGL on LDs.

**Figure 2 biomolecules-12-00057-f002:**
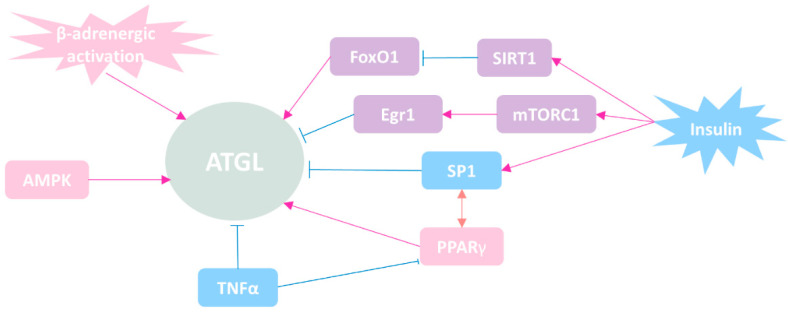
Simplified overview of regulation of ATGL expression. ATGL is demonstrated to be activated by β-adrenergic activation. AMPK phosphorylates ATGL at Ser 406 therefore activates its enzymatic activities. Insulin inhibits ATGL expression through upregulating SIRT1, which restrains the nuclear localization of FoxO1 by deacetylating. Insulin also inhibits ATGL expression through Egr1-mTORC1 signaling pathway. Sp1 has an inhibitory control over ATGL in preadipocytes, while the functional interaction of PPARγ and Sp1 transactivates ATGL in mature adipocytes. TNF-α downregulates ATGL mRNA level but does not alter its protein level.

**Figure 3 biomolecules-12-00057-f003:**
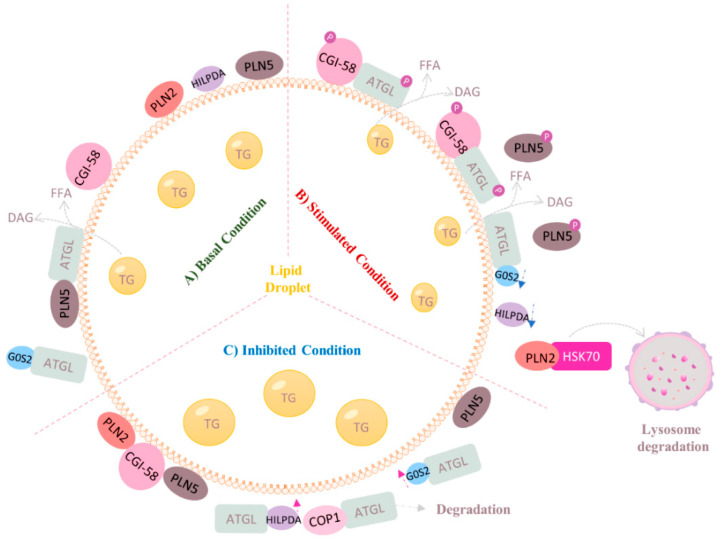
Regulation of ATGL at post-translational level in the liver. (**A**) In basal condition: LDs are coated with PLN2 and PLN5, which restrain the access of ATGL to the stored TGs. PLN5 also directly binds to ATGL, competing against CGI-58 for ATGL interaction. (**B**) In stimulated condition, PKA-mediated phosphorylation of PLN5 results to the release of ATGL and subsequently the co-activation by CGI-58. PLN2 is degraded through chaperone-mediated autophagy, which exposes the surface of LDs to ATGL, therefore stimulating lipolysis. (**C**) In inhibited condition: ATGL inhibitors HILPDA and G0S2 are upregulated, which enhances the inhibitory control over ATGL. E3 ubiquitin ligase COP1-mediated proteasomal degradation reduces the protein levels of ATGL, hence inhibiting lipolysis.

**Figure 4 biomolecules-12-00057-f004:**
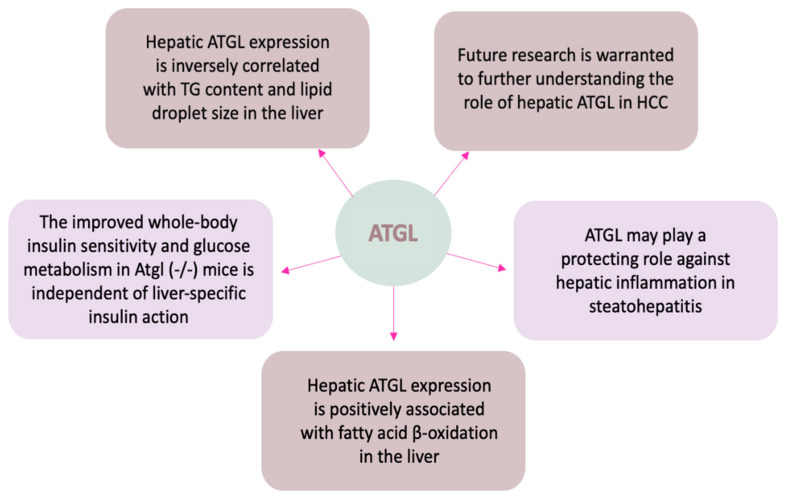
Role of ATGL in regulation of liver pathophysiology. Hepatic ATGL is inversely correlated with TG contents and LD sizes in the liver. Additionally, ATGL-mediated lipolysis provides substrates (FAs) for sustaining FA oxidation and for coordinating the transcriptional program required for FA oxidation. Furthermore, ATGL may play a protective role against inflammatory responses in the liver. Last but not least, although studies have shown that hepatic ATGL is not required for direct systemic glycaemic control, its significance in controlling hepatic and systemic glucose metabolism cannot be excluded considering the mutual coordination of glucose and FAs as major fuels.

**Table 1 biomolecules-12-00057-t001:** Studies of mouse models with global- and liver-specific mutation of ATGL.

Author	Year	Animal Model	Animal Age	Key Findings
Haemmerle et al. [67]	2006	Global ATGL inactivation by targeted homologous recombination	8 to 14 weeks	ATGL is the rate-limiting enzyme of TG catabolism. The inactivation of ATGL increased glucose tolerance and insulin sensitivity.
Reid et al. [83]	2008	Adenovirus-mediated global ATGL knockout in *ob/ob* mice	3 to 5 months old	ATGL possesses TG hydrolase activity and is essential in maintaining hepatic lipid homeostasis by mobilizing and partitioning stored TG into FA oxidation pathways.
Kienesberger et al. [84]	2009	Global ATGL knockout in mice with mixed genetic background (50% C57BL/6 and 50% 129/Ola)	<14 weeks	Global ATGL deficiency decreased insulin signaling in the liver of the mice.
Turpin et al. [85]	2011	Adenovirus-mediated liver-specific ATGL overexpression in *ob/ob* mice	10 weeks	Liver-specific ATGL overexpression reduces hepatic steatosis and mildly enhances liver insulin sensitivity.
Wu et al. [86]	2011	Albumin Cre-mediated liver specific ATGL knockout	6, 8, and 12 months old	ATGLLKO induced hepatic steatosis and suppressed β-oxidation in the liver.
Ong et al. [87]	2011	Adenovirus-mediated liver-specific ATGL knockdown in C57/B16 mice	8 to 10 weeks old	Liver-specific knockdown of ATGL reduced TG hydrolase activity, and increased TG content in the liver. It also altered fatty acid composition with a significant reduction in C16:0, C18:0, and C18:3 but an increase in C18:1 in hepatic TG content.
Fuchs et al. [88]	2012	Global ATGL inactivation by targeted homologous recombination	N/A	The increased non-esterified oleic acid (OA) in the liver protected ATGL KO mice from TM-induced hepatic ER stress through interfering with palmitate (PA)-induced phosphoinositide-3-kinase inhibitor 1 (Pik3ip1) expression.
Jha et al. [89]	2014	Global ATGL inactivation by targeted homologous recombination	7 to 10 weeks	ATGL deficiency enhanced MCD- and LPS-induced hepatic inflammation. The anti-inflammatory effect of ATGL in the liver was partially achieved by PPARα signaling pathway.

## Data Availability

Not applicable.

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
