# Peer review of "Adipose Triglyceride Lipase in Hepatic Physiology and Pathophysiology"

_biomolecules, 2021, doi:10.3390/biom12010057_

Round 1
Reviewer 1 Report
Very nice review showing the crucial role of Adipose triglyceride lipase in hepatic physiology and pathophysiology. However, since hepatic ATGL levels are upregulated upon fasting and patients with NAFLD have reduced hepatic ATGL levels, the relationship between autophagy and ATGL must be addressed.
Reviewer 2 Report
The review is well-curated and covers the importance of ATGL in the context of the Liver. Following queries could be addressed for clarity.
- Since a putative tumor suppressor function is suggested for ATGL, and since there are several studies (Di Leo Oncogene 2019, M Liu JB Mol. Tox. 2019, etc.)that show the importance of ATGL in hepatocellular carcinoma, a paragraph could be added about the importance of ATGL mediated lipolysis in Liver cancer.
- It is worth mentioning that ATGListatin does not inhibit human ATGL. G0S2 peptide-based inhibition of ATGL (Cerk et al. JBC 2014) could be another viable option.
- Most of the lipolysis regulation mechanisms discussed are studied in the context of adipose tissues. The authors should be careful to highlight findings in the Liver.
- It is also worth mentioning the inhibition of ATGL by lipid intermediates such acyl coA, and the role of fatty acid-binding proteins., also hepatic mechanisms such as how hepatic ATGL channels hydrolyzed FAs to β-oxidation and induce PPAR-α signaling( Ong JLR 2014) and follow-up studies if any.
